# Viscoelastic Soil–Structure Interaction Procedure for Building on Footing Foundations Considering Consolidation Settlements

**Ricardo Morais Lanes** [1], **Marcelo Greco** [1,*] and **Valerio da Silva Almeida** [2]

1    Programa de Pós-Graduação em Engenharia de Estruturas, Departamento de Engenharia de Estruturas, Universidade Federal de Minas Gerais, Belo Horizonte 31270-901, Brazil
2    Escola Politécnica, Universidade de São Paulo, São Paulo 05508-220, Brazil
*    Correspondence: mgreco@dees.ufmg.br

**Abstract:** This paper presents a numerical methodology to analyze frame structures supported on footing foundations subjected to slow strains caused by consolidation settlements. A building project on a subsurface layer of soft soil has been analyzed. The Boundary Element Method with the Mindlin fundamental solution has been applied to compute the displacement resulting from the interference between pressure bulbs on the foundation. The rheological Kelvin–Voigt model has also been used for soil–structure interactions. Terzaghi's Theory of Consolidation was used to fit the displacement–time curves. Finally, the rheological model was coupled through an iterative procedure, employing structural non-linear geometric effects. The results are consistent with settlement predicted effects and revealed that the slow distribution of efforts can cause relevant increases in some regions in the structure of the building.

**Keywords:** soil–structure interaction; consolidation; boundary element method; finite element method; Kelvin–Voigt model





## 1. Introduction

Structural projects usually consider that the foundation supporting soil behaves as a non-deformable solid that remains unchanged after loading. However, practice in foundation design indicates that the soil will deform when subjected to loading and this causes disturbances in the structure. The deformations of the foundation are due to the total settlement, which can be defined by adding portions of the immediate settlement and consolidation settlement.

Soil–structure interaction (SSI) studies are important to control the costs and risks of buildings. Meyerhof [1] was the first to propose an SSI study by beams with a bending stiffness equivalent to the stiffness of superstructures. After, numerous researchers studied the immediate settlement behavior on the basis of the elastic theory and various methods have been proposed. For instance, using the conventional Winkler model [2] or by applying (semi) analytical solutions [3] or using the finite layer method, presented in Booker et al. [4]. The SSI problem is still an active field of research with recent papers contemplating topics such as: Winkler spring stiffness for continuum soil [5], methods for calculating the equivalent modulus of elasticity of layered soil [6], numerical modeling for no-linearity soil [7], among others [8–10].

In general, those methods only assume the immediate settlement of soil mass. However, for saturated fine-grained soil, the phenomenon of consolidation leads to a more complex behavior of the soil–structure system caused by slow strain. In those cases, the studies must be based on the theory of consolidation.

Many manual methods are known in the literature to estimate consolidation settlements beneath the foundation, citing one-dimensional methods such as the Skempton–Bjerrum, the Stress Path, or the ordinary Terzaghi's theory of one-dimensional consolidation [11]. Nevertheless, these methods have limited assumptions, and for complex general



application are not enough. Vlladkar et al. [12] presented a three-dimensional viscoelastic finite element formulation to study the frame–foundation–soil interaction, considering time-dependent responses for consolidation and creep. Ai et al. [13] used the Boltzmann viscoelastic model and Biot's consolidation equation to numerical modeling the time-dependent interaction between a superstructure, raft, and a saturated soil. It is noted that the numerical simulation of the soil–foundation–structure interaction regarding long-term behavior represents a topic of interest for solving civil engineering problems [14–22].

The Tower of Pisa, some existing constructions in the capital of Mexico (Mexico City), and some seafront buildings in the Brazilian city of Santos are classical examples of structures built on soil subject to consolidation settlements. As a result of slow soil deformation, those structures exhibited large resulting displacements over a long time [23–25].

The novelty of this paper is the presentation of an analysis methodology for treating the viscoelastic behavior of the soil associated with nonlinear structural effects, using an iterative process for the coupling of finite element (FEM) and boundary element (BEM) methods. FEM is used to simulate the geometrical nonlinearity superstructure. BEM with the Mindlin fundamental solution is applied to compute the displacement resulting from the interference between pressure bulbs on the foundation instead of the Winkler model. The Winkler model neglects the interaction between adjacent springs, so the errors tend to grow on soft soils [26]. For the quasi-static evaluation of displacements and long-time stresses, the Kelvin–Voigt rheological model is parameterized. The application is made in an existing structure, but the problem can also be extended to new structures.

## 2. Materials and Methods

### 2.1. Theory of Settlements

The effects of applied loads on saturated soils can be verified through stiffness changes of the material, caused by the interaction between fluids and solids. An increase of external pressure in saturated soil causes interstitial water to flow between grains due to additional neutral pressure occurring. With pore drainage, neutral pressure is converted into effective pressure on the solid portion. For permeable soils, the time required for a total flow of water is short and the solution to the problem is limited to the analysis of the equilibrium state. In this case, the problem is time-independent. However, for fine-grained soils (such as clays), the time required becomes significant and it is necessary to analyze the time-dependent stress–strain behavior. In soil mechanics, this densification is known as primary consolidation settlement. This process involves drainage, compaction, and stress transfer over time, making primary consolidation a complex phenomenon, as shown by Zhu et al. [27]. Terzaghi [28] started the studies of consolidation under one-dimension and that research was subsequently extended to three-dimensional conditions by Biot [29]. In spite of many developments in this area, Terzaghi's theory of one-dimensional consolidation is the most widely used theoretical description for predicting this process, as shown by Olek [30] and Shi et al. [31].

The one-dimensional differential equation that governs primary consolidation settlement through the neutral pressure dissipation process is expressed by:

$$\frac{\partial u}{\partial t} = c_v \frac{\partial^2 u}{\partial z^2} \tag{1}$$

where $c_v$ is the consolidation coefficient of deformable soil, $t$ is the time instant, $u$ is the pore pressure, and $z$ is the depth coordinate of a point in the soil layer.

The exact solution proposed by Terzaghi is given by:

$$u(z,t) = \sum_{m=0}^{m=\infty} \frac{2\Delta\sigma_v}{M} \cdot sen\left(\frac{Mz}{H_d}\right) e^{-T_v M^2} \tag{2}$$

where $M = 0.5\pi(2m+1)$, $m = 0, 1, 2, 3, \ldots, \infty$, $Hd$ is the drainage path, $\Delta\sigma_v$ is the increase of total vertical stress, and $T_v$ is a time factor defined by:

$$T_v = \frac{c_v \cdot t}{H_d{}^2} \tag{3}$$

The local degree of consolidation, $U_z(z,t)$, is defined by the ratio between dissipated neutral pressure up to time instant $t$ and total neutral pressure caused by loading. The mathematical relation is written as:

$$U_z(z,t) = 1 - \frac{u(z,t) - u(z,t_\infty)}{u(z,t_0) - u(z,t_\infty)} \tag{4}$$

where $t_0$ and $t_\infty$ are the initial and final time instants of the analysis.

The average degree of consolidation $U(t)$ is used for the whole soil layer at a given time $t$. It can be obtained by integration of Equation (4) along the depth $z$. This parameter $U(t)$ indicates a settlement percentage reached by the soil layer for a specific time instant $t$.

$$U(t) = \frac{1}{2H_d} \int_0^2 U_z(z,t)dz = 1 - \sum_{m=0}^{m=\infty} \frac{2}{M^2} \cdot e^{-T_v M^2} \tag{5}$$

Taylor [32] proposes empirical Equations (6) and (7) to represent the time factor $T_v$ in terms of the average consolidation parameter $U(t)$.

$$T_v = \frac{\pi}{4}[U(t)]^2 \text{ for } U(t) < 0.6, \tag{6}$$

$$T_v = -0.933\log[1 - U(t)] - 0.085 \text{ for } U(t) > 0.6 \tag{7}$$

The valid hypotheses for applying the equations described above are the isotropic behavior of soil, complete drainage at the bottom and top of the deformable layer, and neglected soil-specific gravity.

In addition to primary consolidation, the occurrence of secondary consolidation settlement, also known as creep, should also be highlighted. This mechanism always occurs after primary consolidation, after the neutral pressure has almost completely dissipated, and considering the soil under a scenario of constant effective stress over time. For creep modeling, a three-parameter model such as Boltzmann [33] and a four-parameter model such as Burgers [34] are recommended. In most soils, creep is less important, because its magnitude is lower than in other types of settlement. For this reason, it is not considered in most of the analyses. Settlement measurements carried out over many years on structures allow for classifying soils in terms of creep. Regarding sand settlements, it is practically non-existent. In clays, it is common to assume a small portion of 3 to 10% of the total settlement by consolidation, as shown by Cosenza and Korosak [35]. In the present study, intended for building structures, creep settlements will be neglected since normally these structures need to meet durability conditions for around 50 years.

The classical mechanism of settlement in saturated soils assumes that the soil particles and the pore water are incompressible; changes in volume must be due to changes in void ratio as the water flows out of (or into) the soil. Therefore, the volumetric variation is null at the beginning of loading. However, in unsaturated soils, immediate settlements are manifested at the initial moment before the consolidation mechanism. Instant settlement can be treated numerically with the application of the Theory of Elasticity, in which the behavior of the material is simply represented by the Young modulus ($E$) and Poisson ratio ($v$) or even by more elaborate numerical methods, considering the elastoplastic behavior of materials.

### 2.2. Coefficient of Consolidation

The coefficient of consolidation ($c_v$) determines the speed of excessive neutral pressure dissipation. This parameter is usually determined from the vertical displacement evolution of a soil sample along the time for each loading stage. The most adopted methods to determine the coefficient of consolidation are: the square root of time, as shown by Taylor [32] and the logarithm time method, as shown by Casagrande [36], both developed from data fitting of consolidation experimental results. These methods aim to adjust experimental results to the theoretical solutions.

An empiric correlation between the coefficient of consolidation and other physical indexes was presented by Carrier [37], as follows:

$$c_v = \frac{28.67}{PI} \cdot \frac{\left(1.192 + A_c^{-1}\right)^{6.993} \cdot \left(4.135 \cdot LI + 1\right)^{4.29}}{\left(2.03IL + 1.192 + A_c^{-1}\right)^{7.993}} \left(\frac{\text{m}^2}{\text{year}}\right) \tag{8}$$

where $PI$ is the plasticity index, $LI$ is the liquidity index, and $A_c = \frac{PI}{CF}$ is the colloidal activity of clay, with $CF$ being equal to clay size fraction (percentage of material with granulometric size littler than 2 $\mu$m).

Although a constant coefficient of consolidation is usually assumed, it is known that the void ratio and the compressibility and permeability coefficients vary during compression, resulting in an effective variation of the parameter. However, it is established that, in general, these variations do not affect the results too excessively, as shown by Pinto [38] and Spannenberg [39].

Figure 1 shows the variation range of the consolidation coefficient to clays typical of the Rio de Janeiro region, obtained from more than one hundred oedometric tests performed by several researchers, as shown by Almeida et al. [40], Ortigão [41], Formigheri [42], Sayão [43], and Spannenberg [39]. It is observed that the dispersion of results is very large for the interval where the material is subject to effective pressures up to 100 kPa. For values in which the effective pressure exceeds 100 kPa, the results comprise the range of $0.5 \pm 0.3$ m$^2$/year.

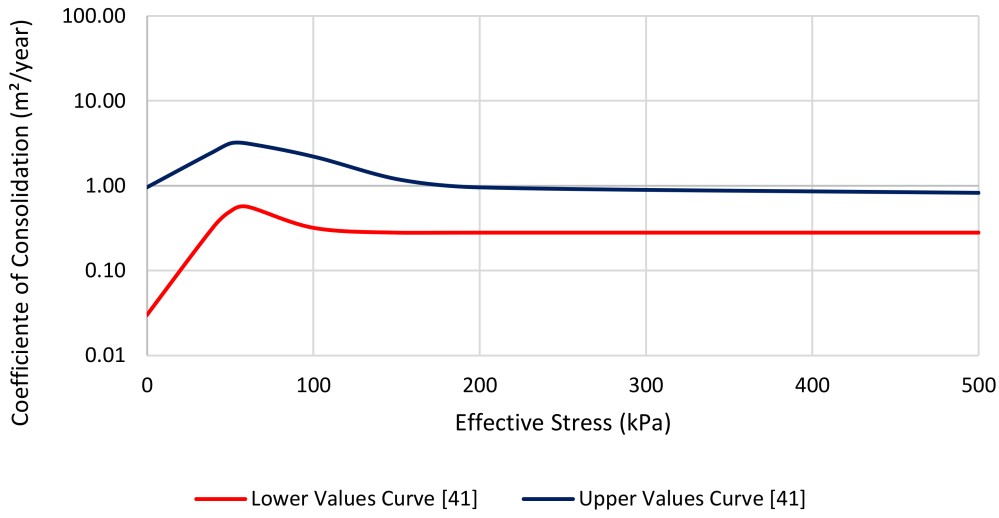

**Figure 1.** Range of coefficient of consolidation values for clays in the region of Rio de Janeiro obtained from oedometric tests [39–43].

### 2.3. Kelvin–Voigt Model Applied to Boundary Element Formulation

Most of the problems in geomechanics have an extensive list of references by applying the Boundary Element Method (BEM), as seen in Witt [2], Davies and Banerjee [3], and Ai et al. [13,14,20]. The numerical procedures such as the use of Finite Element (FEM), Finite Difference (FDM), or Boundary Element (BEM) methods have been applied extensively

for more than 40 years for solving a three-dimensional continuity medium. One of the advantages of the BEM is the reduction of the problem dimension by one and the implicit fulfillment of the radiation condition for unbounded domains, which makes the BEM usually prefarable for the calculation of infinite or semi-infinite domains, e.g., soils [44].

The problem of one-dimensional consolidation behavior of viscoelastic soils is also a current research topic [45]. Huang [45] uses the Caputo–Fabrizio fractional derivative to estimate the viscoelastic properties of soils, and further the one-dimensional consolidation equation is derived to simulate the consolidation behavior of soils. The authors state that a higher viscosity coefficient of the Kelvin–Voigt body delivers the faster dissipation of excess pore–water pressure in the early stage, but a slower one at a later stage.

The theory of viscoelasticity states that viscoelastic materials loaded by constant or variable forces present states of stress and are strain-dependent on time-varying. The Kelvin–Voigt viscoelastic model depends on two parameters (one elastic and another related to damping) and allows the strain-description of soils subjected to settlements due to consolidation, as shown by Wang et al. [46] and Huang and Li [47].

Alipour and Rajabi [48] considered a viscoelastic substrate based on the Kelvin–Voigt model to describe the soil behavior, which is used to reduce base vibrations. Other viscoelastic models can be used for several applications. For instance, Riobom Neto et al. [33] used the Boltzmann model in a formulation based on the BEM applied to in-plane and infinite half-plane viscoelastic quasi-static problems considering only shear effects. Oliveira et al. [49] investigate the two-dimensional structural behavior of structures using the Boltzmann model for creep analysis. The shear effects on a concrete viaduct segment are modeled. Thus, it appears that simpler models are still a good alternative for modeling viscoelastic problems. The main difficulty to model footing foundations using boundary elements is related to differential settlements. The present work proposes a methodology to consider this kind of severe situation.

The unidimensional Kelvin–Voigt model is an arrangement of a spring (elastic element) with a constant modulus of elasticity $E$ and a damper (viscous element) with a viscosity constant $\eta$ (Figure 2).

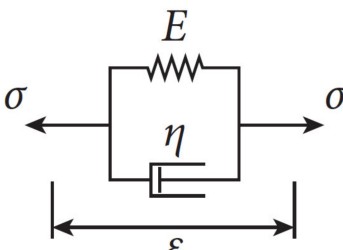

**Figure 2.** Representation of unidimensional Kelvin–Voigt model.

In the Kelvin–Voigt model, the strain is the same for both the spring and the damper. Thus, total strain ($\varepsilon_{lm}$) is equal to elastic strain ($\varepsilon_{lm}^{e}$) and equivalent viscous strain ($\varepsilon_{lm}^{v}$), as follows:

$$\varepsilon_{ij} = \varepsilon_{lm}^{e} = \varepsilon_{lm}^{v} \tag{9}$$

The resulting stress ($\sigma_{ij}$) in the model can be given by the sum of stress in the elastic element ($\sigma_{ij}^{e}$) and stress in the viscous element ($\sigma_{ij}^{v}$), expressed by:

$$\sigma_{ij} = \sigma_{ij}^{e} + \sigma_{ij}^{v} \tag{10}$$

where $\sigma_{ij}^{e} = C_{ij}^{lm}\varepsilon_{lm}^{e}$ and $\sigma_{ij}^{v} = \eta_{ij}^{lm}\dot{\varepsilon}_{lm}^{v}$.

It should be noted that viscous stress is proportional to strain rate ($\dot{\varepsilon}_{lm}^{e}$), considering $C_{ij}^{lm}$ as the constitutive tensor and $\eta_{ij}^{lm}$ as the viscous tensor, defined according to Mesquita and Coda [50] as:

$$C_{ij}^{lm} = \lambda \cdot \delta_{ij} \cdot \delta_{lm} + \mu \cdot \left( \delta_{ij} \cdot \delta_{jm} + \delta_{im} \cdot \delta_{jl} \right), \tag{11}$$

$$\eta_{ij}^{lm} = \theta_\lambda \cdot \lambda \cdot \delta_{ij} \cdot \delta_{lm} + \theta_\mu \cdot \mu \cdot \left(\delta_{ij} \cdot \delta_{jm} + \delta_{im} \cdot \delta_{jl}\right) \tag{12}$$

For plane stress states, the viscous tensor can be written as:

$$\eta_{ij}^{lm} = \frac{2\theta_\mu\mu}{\theta_\lambda\lambda + 2\theta_\mu\mu} \cdot \begin{bmatrix} 2\cdot(\theta_\lambda\cdot\lambda + \theta_\mu\cdot\mu) & \theta_\lambda\cdot\lambda & 0 \\ \theta_\lambda\cdot\lambda & 2\cdot(\theta_\lambda\cdot\lambda + \theta_\mu\cdot\mu) & 0 \\ 0 & 0 & \theta_\lambda\cdot\lambda + 2\cdot\theta_\mu\cdot\mu \end{bmatrix} \tag{13}$$

where $\theta_\lambda$ and $\theta_\mu$ are viscous parameters of the material. According to Mesquita and Coda [50], these parameters can be obtained from a uniaxial tension test and shear test. The Lamé constants are classically expressed by:

$$\lambda = \frac{\nu\cdot E}{(1+\nu)\cdot(1-2\nu)} \text{ and } \mu = G = \frac{E}{2(1+\nu)} \tag{14}$$

In the majority of materials, the viscous tensor can be presented in a simpler form, depending only on a single viscous parameter. In this case, one has $\theta_\lambda = \theta_\mu = \gamma$ and consequently Equation (12) becomes simpler, as follows:

$$\eta_{ij}^{lm} = \gamma\left[\lambda\cdot\delta_{ij}\cdot\delta_{lm} + \mu\cdot\left(\delta_{ij}\cdot\delta_{jm} + \delta_{im}\cdot\delta_{jl}\right)\right] = \gamma C_{ij}^{lm} \tag{15}$$

Equation (10) can be rewritten in terms of elastic strains and strain rates.

$$\sigma_{ij} = C_{ij}^{lm}\cdot\varepsilon_{lm} + \eta_{ij}^{lm}\cdot\dot{\varepsilon}_{lm} \tag{16}$$

$$\sigma_{ij} = C_{ij}^{lm}\cdot\varepsilon_{lm} + \gamma\cdot C_{ij}^{lm}\cdot\dot{\varepsilon}_{lm} \tag{17}$$

for $\forall\, \theta_\lambda = \theta_\mu = \gamma$.

Boundary element formulation based on the Kelvin–Voigt model is obtained from a weighted residues technique. The integral equilibrium equation is given by:

$$\int_\Gamma u^*_{ki}\cdot p_i\cdot d\Gamma - \int_\Omega \sigma^*_{klm}\cdot\varepsilon_{lm}\cdot d\Omega + \int_\Omega u^*_{ki}\cdot b_j\cdot d\Omega = 0 \tag{18}$$

where the terms $u^*_{ki}$ and $\sigma^*_{klm}$ are related to the fundamental solution of the equation, and $\Gamma$ and $\Omega$ represent the boundary (bidimensional) and domain (tridimensional).

Considering the elastic relation $\sigma^*_{klm}\cdot\varepsilon_{lm} = \varepsilon^*_{kij}\cdot C_{ijlm}\varepsilon_{lm}$, Equation (18) can be rewritten as follows:

$$\sigma^*_{klm}\cdot\varepsilon_{lm} = \varepsilon^*_{kij}\cdot C_{ij}^{lm}\cdot\varepsilon_{lm} + \varepsilon^*_{kij}\cdot\gamma\cdot C_{ij}^{lm}\cdot\dot{\varepsilon}_{lm} \tag{19}$$

$$\int_\Gamma u^*_{ki}\cdot p_i\cdot d\Gamma - \int_\Omega \varepsilon^*_{kij}\cdot C_{ij}^{lm}\cdot\varepsilon_{lm}d\Omega - \int_\Omega \varepsilon^*_{kij}\cdot\gamma C_{ij}^{lm}\cdot\dot{\varepsilon}_{lm}d\Omega + \int_\Omega u^*_{ki}\cdot b_i d\Omega = 0 \tag{20}$$

The simplest rheological model considers only one viscous parameter ($\theta_\lambda = \theta_\mu = \gamma$). This model was chosen because it facilitates the transformation of domain integrals into boundary integrals.

$$\varepsilon^*_{kij}\gamma C_{ij}^{lm}\dot{\varepsilon}_{lm} = \gamma\sigma^*_{klm}\dot{\varepsilon}_{lm} = \gamma\sigma^*_{klm}\dot{u}_{l,m} = \gamma\sigma^*_{kij}\dot{u}_{i,j} \tag{21}$$

Equation (20) can be rewritten as follows:

$$\int_\Gamma u^*_{ki}p_i\,d\Gamma - \int_\Omega \sigma^*_{kij}u_{i,j}d\Omega - \gamma\int_\Omega \sigma^*_{kij}\dot{u}_{i,j}d\Omega + \int_\Omega u^*_{ki}b_i d\Omega = 0 \tag{22}$$

Applying the divergence theorem and considering integration by parts in second and third terms, one has:

$$\int_\Gamma u_{ki}^* p_i\, d\Gamma - \int_\Gamma \sigma_{kij}^* n_j\, u_i d\Gamma + \int_\Omega \sigma_{kij,j}^* u_i d\Omega - \gamma \int_\Gamma \sigma_{kij}^* n_j \dot{u}_i d\Gamma + \gamma \int_\Omega \sigma_{kij,j}^* \dot{u}_i\, d\Omega + \int_\Omega u_{ki}^* b_i\, d\Omega = 0 \tag{23}$$

Considering the tractions definition ($p_{ki}^* = \sigma_{kij}^* n_j$) and Dirac Delta function in Equation (23), one has:

$$\overline{C}_{ki} u_i(p) + \gamma \overline{C}_{ki} \dot{u}_i(p) = \int_\Gamma u_{ki}^* p_i d\Gamma - \int_\Gamma p_{ki}^* u_i d\Gamma - \gamma \int_\Gamma p_{ki}^* \dot{u}_i\, d\Gamma + \int_\Omega u_{ki}^* b_i\, d\Omega \tag{24}$$

To obtain the system of equations for the 3D semi-infinite half-plane, based on Equation (24), it is necessary to consider the Mindlin fundamental solution. The complete fundamental solution can be found in Brebbia et al. [51].

Implementations of the formulations presented in Section 2 were performed using the MatLab programming language.

## 3. Results

### 3.1. Problem Description

Cavalcanti et al. [52] evaluated the damage caused to a 26-story building built on footings, raised in the 1980s in the metropolitan region of Recife, Brazil. The anomalies and displacements of the construction were monitored over 10 years. After this period, it was found that the displacements did not indicate a stabilization trend, with a settlement speed of approximately 10 mm/year. A reinforcement of the foundation was proposed using piles.

The present case study was inspired by the publication by Cavalcanti et al. [52]. Here, simplifications were conducted for the geometrics of the building structure, but foundations and soil information were maintained. Here, the geometrical nonlinear effects in the building were also considered, as well as the viscoelastic behavior of the soil for showing the methodology.

The fictitious building under analysis is a hotel with 19 storeys (with a ceiling height of 300 cm) supported by shallow foundations. The geometric layouts of a standard storey and ribbed slab are shown in Figure 3a. The building foundation is demonstrated in Figure 3b, with lengths in the plan by heights of each footing. The floors of the building were designed to support their weight, average accidental load of 1.75 kN/m$^2$, and an additional permanent load of 8.40 kN/m$^2$ (masonry and coatings).

Admitting the same behaviors as the displacements in the soil mass, as shown in Cavalcanti et al. [52], the average settlement speed of the structure is equal to 10 mm/year in that example. That speed of settlement also presents alike values that were observed in the Brazilian city of Santos [24,25], between 10 and 15 mm/year. The typical local drilling profile is shown in Figure 4, where a thick layer of clay-silt material with low-strength organic matter can be seen, located about 10 m deep from the footing's settlement quota. The water table level appears at $-4.0$ m elevation. The average working stresses of footing, acting before the occurrence of consolidation, are presented in Table 1. The effects of wind were not considered.

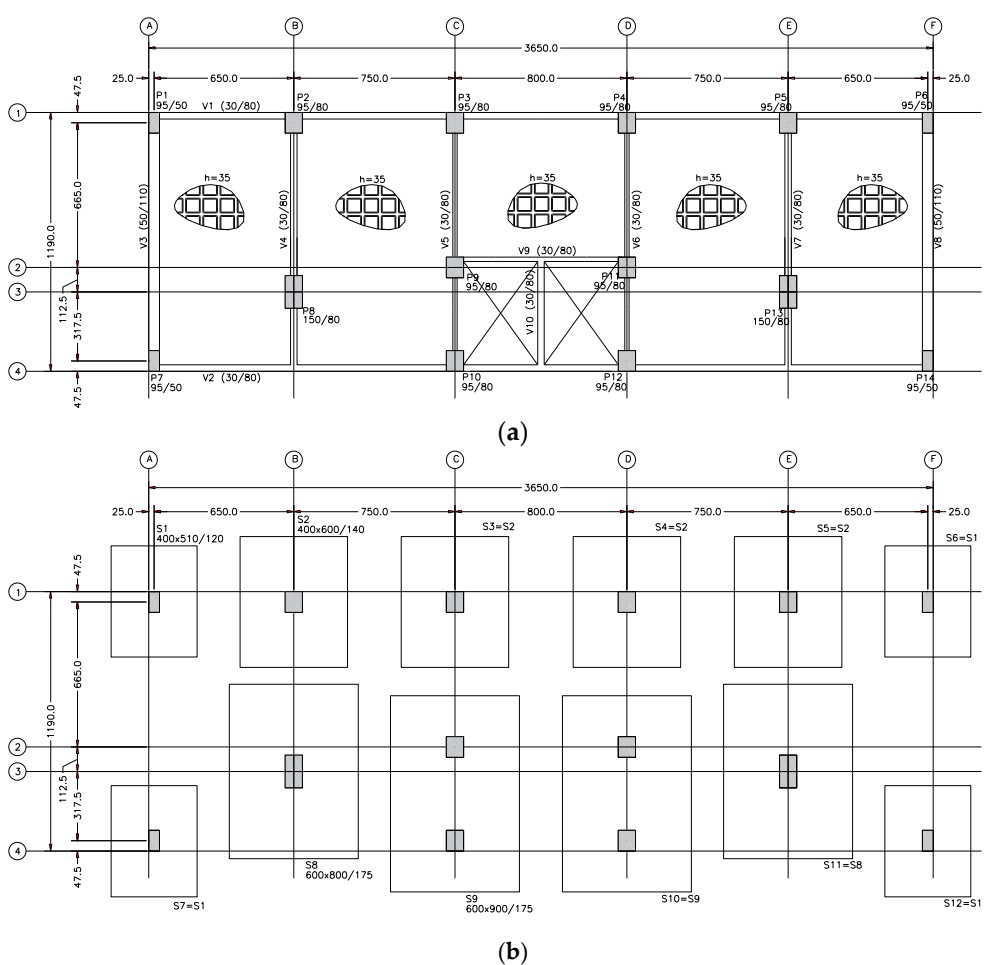

(**a**)

(**b**)

**Figure 3.** Building structural design (cm): (**a**) standard storey; (**b**) foundations.

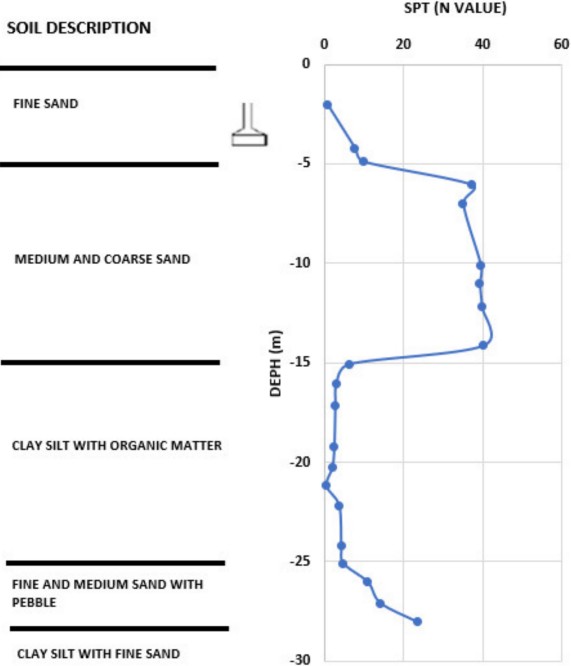

**Figure 4.** Surveys carried out during the building design phase [38].

**Table 1.** Vertical normal stresses applied for each footing.

| Footing | S1 = S6 | S2 = S5 | S3 = S4 | S7 = S12 | S8 = S11 | S9 = S10 |
|---|---|---|---|---|---|---|
| Dimensions (m) | 4.0 × 5.1 | 5.0 × 6.0 | 5.0 × 6.0 | 4.0 × 5.1 | 6.0 × 8.0 | 6.0 × 8.0 |
| Area (m$^2$) | 20.4 | 30.0 | 30.0 | 20.4 | 48.0 | 48.0 |
| Average vertical stress (kPa) | 332.0 | 356.0 | 337.0 | 360.0 | 385.0 | 357.0 |

*3.2. Interaction Soil–Foundation*

In that paper, the approximated relationship between the coefficient of consolidation and the effective stress was used, as indicated in Figure 1. The average effective stress from the building at the center of the deformable soil layer is approximately 250 kPa, which results in an average coefficient of consolidation value equal to 0.5 m$^2$/year, as shown in Figure 1. Alternatively, from laboratory tests, it is possible to specify the coefficient of consolidation for the deformable soil layer in Figure 4, using the square root of time, logarithm time method, or the universal empiric correlation given by Equation (8), and applying Terzaghi's theory of one-dimensional consolidation.

As the field monitoring indicated the absence of stability of movements, it is desired to identify the time required for the stabilization of settlements by consolidation. For this purpose, the equations that allow for relating the time factor to the average degree of consolidation are used. That paper assumes two drainage boundaries, one upper and one lower of the layer deformable soil.

Equation (3) offers the time factor, $T_v = 0.2$, assuming $t = 10$ years, $c_v = 0.5$ m$^2$/year, and $Hd = 5.0$ m. Applying $T_v = 0.2$ at Equation (6), it indicates $U(t) = 50.4\%$.

The average degree of consolidation, $U(t)$, of 90.0% is given by Equation (7) and offers $T_v = 0.848$. Applying $T_v = 0.848$, $Hd = 5.0$ m, and $c_v = 0.5$ m$^2$/year in Equation (3), time $t = 42.4$ years is obtained. Thus, the average total displacement for $U(t) = 100\%$ reaches approximately 100 mm × 1.000/0.504~198 mm at infinite time. The development of vertical displacements for soil mass over time is shown in Figure 5.

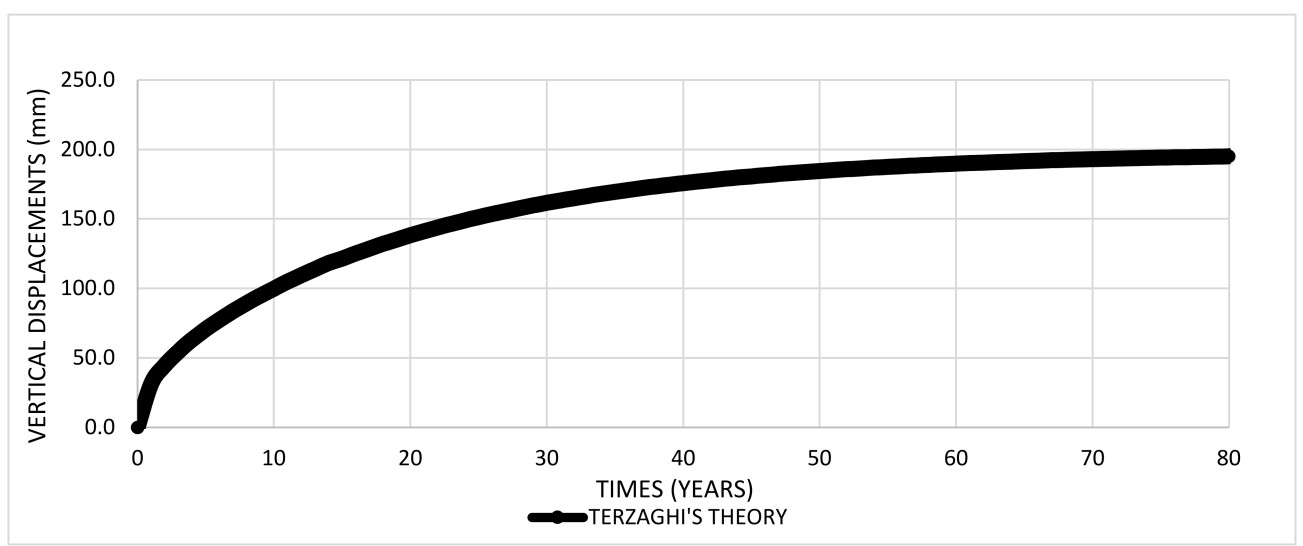

**Figure 5.** Expected displacement curve for soil mass over time.

With the results of Terzaghi's theory of one-dimensional consolidation for the soil mass, it is assumed that the displacement versus time curve of footings and columns of the structure lies around the curve that describes the theory of consolidation for the entire local soil mass. The footings model is simulated using boundary elements and the Kelvin–Voigt viscoelastic model to fit the curves and identify the soil parameters.

The proposed methodology for this curve fitting consists of simulating the boundary element model with attributions of vertical stresses corresponding to each footing at the

initial time instant $t = 0$. The vertical stresses of the initial instant and the numerical model of the foundation are presented, respectively, in Table 1 and Figure 6. Note that the BEM model shown in Figure 6 allows for consideration of the influence of one footing on the others.

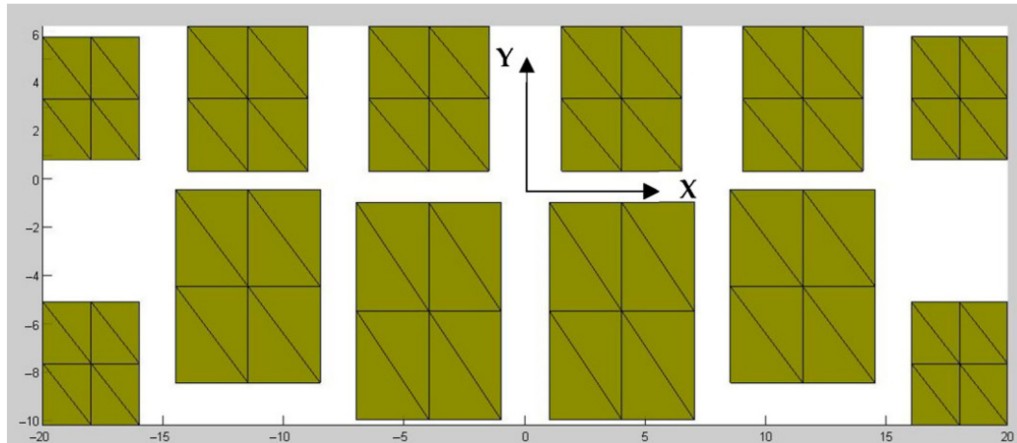

**Figure 6.** Boundary elements mesh with 96 triangular elements (constant strain approximation) used to analyze the footings.

Although there is a redistribution of stresses between the footings over time, the stresses presented in Table 1 result in a constant vertical load on the soil mass, which validates the application of Kelvin–Voigt through boundary elements to define the geotechnical parameters. With the value of the Poisson ratio of the soil ($v \cong 0.35$) and arbitrated Young modulus ($E$), the shear modulus, $G$ presented in Equation (14) can be calculated. Then, it is possible to obtain the average displacements of all the footings for infinite time, when the variable strain rate is null, $\varepsilon_{lm} = 0$, in the Kelvin–Voigt model.

The ratio between the final average displacements of all footings and of the soil mass in Terzaghi's theory, multiplied by the arbitrated Young modulus, gives the equivalent Young modulus which corresponds to a curve fit. The procedure makes the originally viscoelastic problem into an equivalent elastic one, depending only on the stresses, given a fixed time.

The soil viscosity factor ($\gamma$) is determined iteratively until the displacement at a given instant of time is compatible with the displacement predicted by the consolidation theory. In the present work, $\gamma$ was defined for the average displacements of the footings equal to 100 mm in the time $t = 10$ years.

After the described procedures, $v = 0.35$, $E = 24{,}221.0$ kPa, $G = 8970.0$ kPa, and $\gamma = 14.0$ years are obtained for the foundation soil mass in a viscous regime. The development of the expected displacements for the footings, under the stress hypothesis at $t = 0$, is shown in Figure 7. It is noted that the weighted average displacements of the footings converge to the displacement predicted for the soil mass by Terzaghi's theory in all moments.

### 3.3. Interaction Foundation–Structure

Once the viscoelastic parameters of the soil mass are known, it is possible to perform the soil–foundation–structure coupling to the infinite time. This procedure transforms a viscoelastic study into a trivial elastic study at a specific time instant. The foundation is simulated using the BEM, while the superstructure is modeled using the FEM with nonlinear geometric effects. The coupling of models in the BEM and FEM is an iterative process.

Comparing the considered methodology with the conventional soil modeled by finite elements, it is possible to state that here the computational cost is considerably lower. For instance, Khosravifardshirazi et al. [53] have analyzed a 10-storey concrete structure under

a slab foundation coupled with an elastic soil using a classical finite elements approach and used around 2 million 3D solid elements to model the problem.

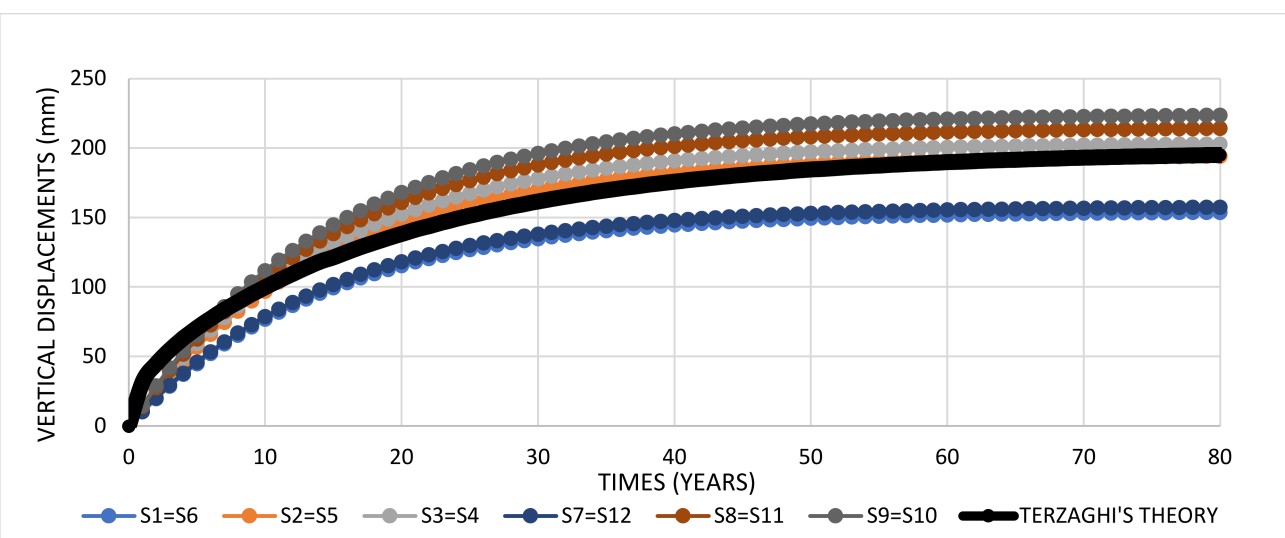

**Figure 7.** Vertical displacements on footings, with soil–foundation interaction, obtained for Terzaghi's theory fitting at $t = 0$.

The proposed methodology consists of calculating the support reactions of the structure through the FEM, initially considering the foundation as rigid supports [54]. Then, from reactions resulting in the superstructure model, the settlements of all footings are determined using the foundations model where they are obtained via the BEM and used to determine the spring coefficient for simulating an elastic soil condition.

In a new stress analysis, rigid supports are replaced by spring coefficients so that new spring coefficients are obtained from the results found in the superstructure and foundation models (Winkler model). The iterative process ends when the spring coefficients, support reactions, or displacements converge to the same value. The finite elements modeling of the superstructure model is presented in Figure 8.

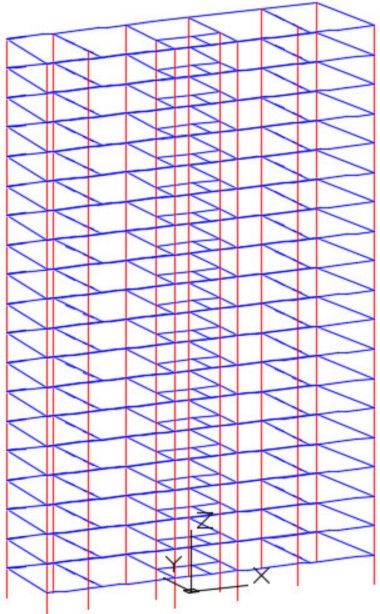

**Figure 8.** Finite elements mesh with beam elements (in blue) and columns (in red) used to simulate the superstructure of a 19-storey building.

The results of spring stiffness, vertical forces, and displacements of footings are presented, respectively, in Tables 2–4 for all iterations of the BEM/FEM coupling. It is observed that five iterations were necessary for convergence of the displacement with errors less than 2%, considered acceptable for this type of analysis.

**Table 2.** Spring stiffness evolution in the FEM model (kN/m).

| Iteration | S1 = S6 | S2 = S5 | S3 = S4 | S7 = S12 | S8 = S11 | S9 = S10 |
|-----------|---------|---------|---------|----------|----------|----------|
| 1 | Infinite | Infinite | Infinite | Infinite | Infinite | Infinite |
| 2 | 44,013 | 55,031 | 49,768 | 46,430 | 86,379 | 76,402 |
| 3 | 48,814 | 53,420 | 47,841 | 53,304 | 83,166 | 73,317 |
| 4 | 51,179 | 52,021 | 45,009 | 55,457 | 80,062 | 71,199 |
| 5 | 53,512 | 52,974 | 47,163 | 57,189 | 79,728 | 72,563 |

**Table 3.** Footing reactions evolution in the FEM model (kN).

| Iteration | S1 = S6 | S2 = S5 | S3 = S4 | S7 = S12 | S8 = S11 | S9 = S10 |
|-----------|---------|---------|---------|----------|----------|----------|
| 1 | 6778 | 10,676 | 10,103 | 7336 | 18,485 | 17,114 |
| 2 | 7859 | 10,310 | 9616 | 8955 | 17,548 | 16,203 |
| 3 | 8598 | 10,144 | 9497 | 9705 | 16,813 | 15,735 |
| 4 | 9097 | 10,118 | 9244 | 10,008 | 16,424 | 15,601 |
| 5 | 9151 | 9974 | 9361 | 10,176 | 16,148 | 15,682 |

**Table 4.** Displacements evolution on footings in the BEM model (mm).

| Iteration | S1 = S6 | S2 = S5 | S3 = S4 | S7 = S12 | S8 = S11 | S9 = S10 |
|-----------|---------|---------|---------|----------|----------|----------|
| 1 | 154 | 194 | 203 | 158 | 214 | 224 |
| 2 | 161 | 193 | 201 | 168 | 211 | 221 |
| 3 | 168 | 195 | 211 | 175 | 210 | 221 |
| 4 | 170 | 191 | 196 | 175 | 206 | 215 |
| 5 | 171 | 191 | 198 | 177 | 206 | 219 |

The foundation development of displacements in infinite time after the iteration process for determining parameters is presented in Figure 9. It is noted that, similarly to Figure 6, the weighted average displacements of the footings converge to the displacement predicted for the soil mass by Terzaghi's theory in all moments too. It should be noted that the displacement of S1 = S6 and S7 = S12 foundations are higher than that presented in Figure 6, while the translations of the other foundations are smaller due to the transfer of loads.

As shown in Table 3, the increase of compression load exceeds 35% of the initial value at S1 = S6 and S7 = S12. The internal footings from S2 to S5 and from S8 to S11 indicated a load decrease. That is a result of a transfer of load from footings with more vertical displacements to that with fewer. As the footing dimensions are kept constant, the increase in compression exceeds the allowable stresses of the bases. Assuming theoretical or semi-empirical methods [55] to determine the allowable stress in the footings, these values could be conservatively adopted equal to 400 kPa, while the stress acting on S1 = S6 and S7 = S12 reaches approximately 449 kPa and 499 kPa, respectively, at infinite time. By the elastic linear behavior and the Kelvin–Voigt relationship for soil mass at Figure 9, the stress acting on S1 = S6 and S7 = S12 is equal to 400 kPa at times of 46.6 years and 23.2 years, respectively. Then, interventions must occur at the foundation structure before the age of 23.3 years to avoid its collapse.

The maximum relative rotations between the S1 and S9 footings under vertical loads reaches approximately 1/340. According to EN 1997-1 [56], the maximum value acceptable for buildings range from 1/2000 to about 1/300. For preventing some service limit state from occurring in the structure, a maximum relative rotation of 1/500 is acceptable. Then,

the observed relative rotations may be minimally detrimental to the usefulness of the structure at infinite time.

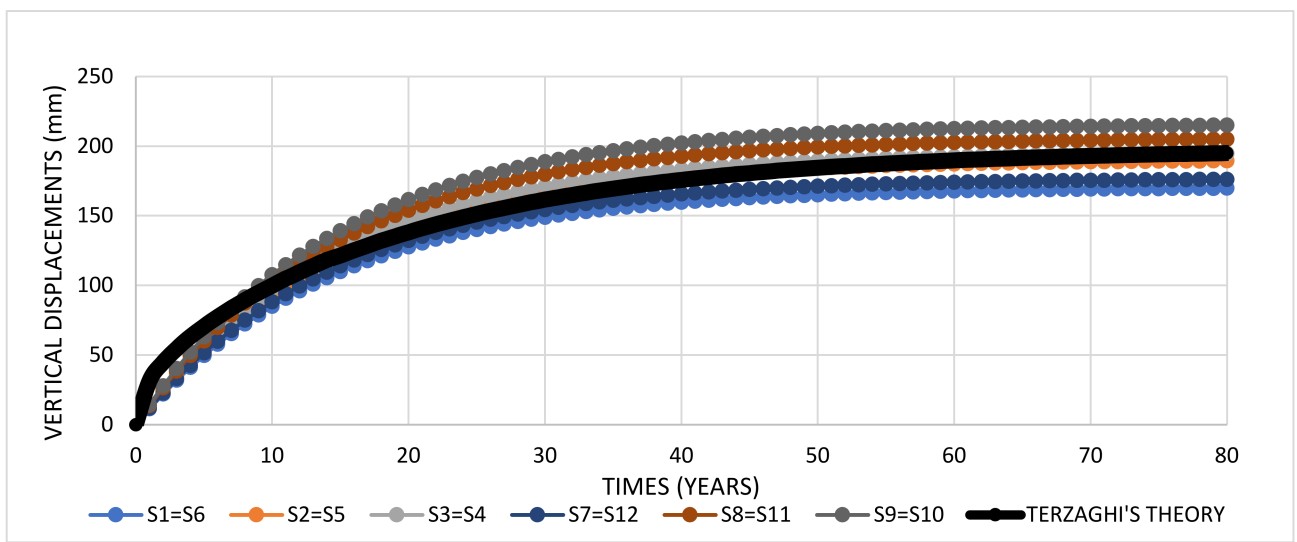

**Figure 9.** Vertical displacements on footings, for the BEM/FEM coupling, at infinite time.

## 4. Discussions about the Superstructure

The redistribution of loads on the foundation is only possible with the reorganization of efforts by the structural elements. The results of the displacements that occurred in the building are discussed below, in addition to the evolution of forces required in the first level of column P7 and the beams V2 and V3 of the standard storey.

The efforts of column P7 (Figure 3) applied on the foundation presented higher force increases due to slow deformation. Table 5 shows that compression forces reach 39% above the initial value. The bending moments increased around 4.6 and 1.5 times the initial value at planes XZ and YZ, respectively.

**Table 5.** Efforts at the first level of column P7.

| Initial Time Results | | | Infinite Time Results after Interactions | | |
|---|---|---|---|---|---|
| Compression Force (kN) | The Maximum Bending Moment at Plane XZ (kN m) | The Maximum Bending Moment at Plane YZ (kN m) | Compression Force (kN) | The Maximum Bending Moment at Plane XZ (kN m) | The Maximum Bending Moment at Plane YZ (kN m) |
| 7336.13 | 56.99 | 184.03 | 10,176.04 | 321.83 | 453.17 |

The beam V2 (Figure 3) is located at the XZ plane. After the excessive neutral pressure dissipation, the maximum shear force reaches 213.3 kN at the support of Axle A. That value is around three times the initial value at the same location. Equally, the bending moment undergoes relevant changes, where a section of Axle A has a value of 705.3 kN m at infinite time, while at the same section it showed 126.8 kN m at initial time. That change represents a difference around of 4.6 times the initial effort (Figures 10 and 11). Axes A to F in Figures 10 and 11 represent reference axes, as shown in Figure 3.

The significant increase in bending moment and shear force at the ends of the beam V2 is consistent with the increase in stiffness of footings S7 and S12 compared to footings S8 to S11, based on the second and fifth iterations of Table 2.

The beam V3 (Figure 3) is located at the YZ plane. There is an increase in the shear force and in the bending moment at the support of Axle 4 and a reduction in those efforts in the support of Axis 1 over time (Figures 12 and 13). Axes 1 to 4 in Figures 12 and 13 represent reference axes, as shown in Figure 3.

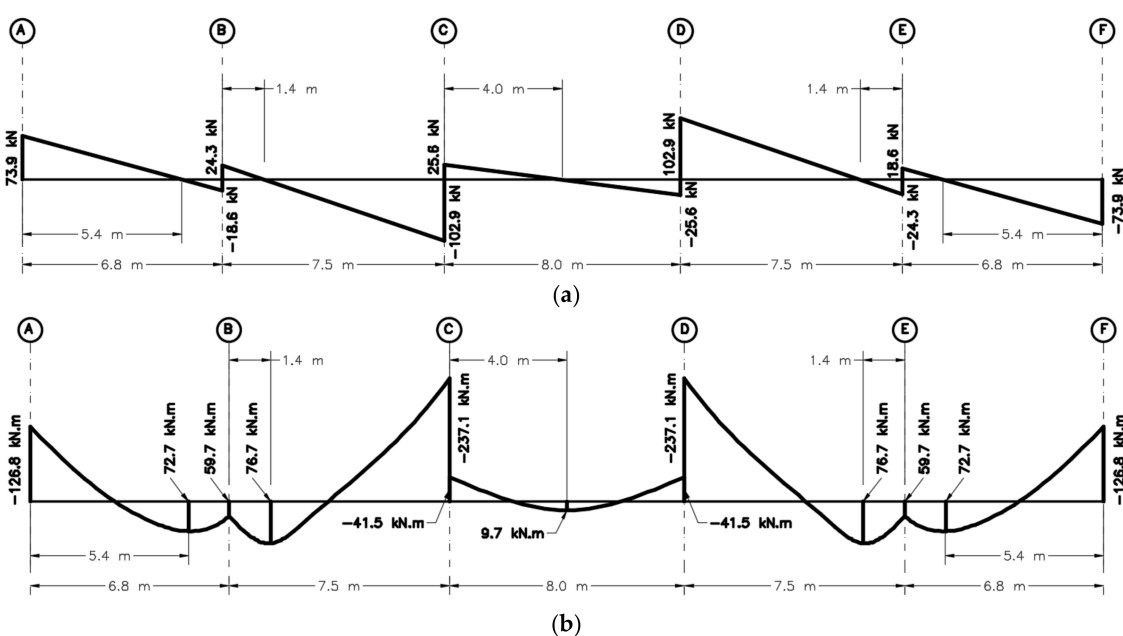

**Figure 10.** Efforts in beam V2 to the initial time instant *t* = 0: (**a**) shear force (kN); (**b**) bending moment (kN m).

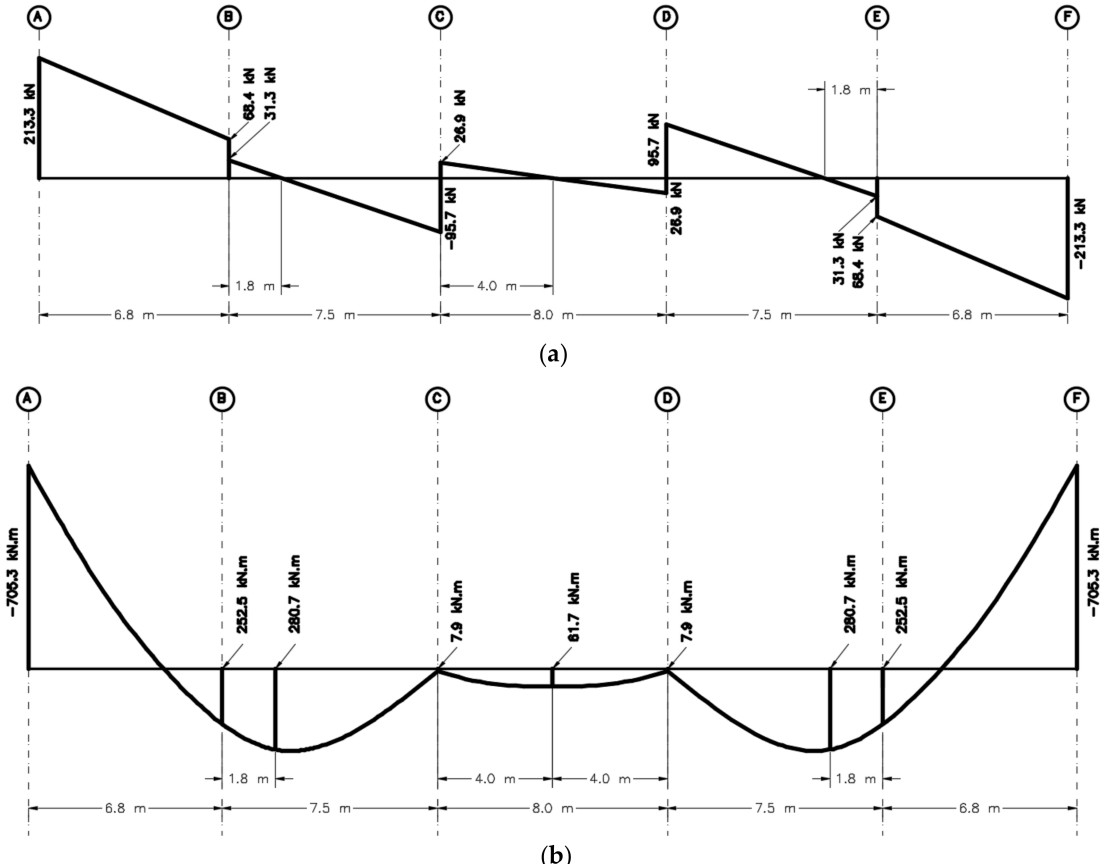

**Figure 11.** Efforts in beam V2 to the infinite time, after iterative process: (**a**) shear force (kN); (**b**) bending moment (kN m).

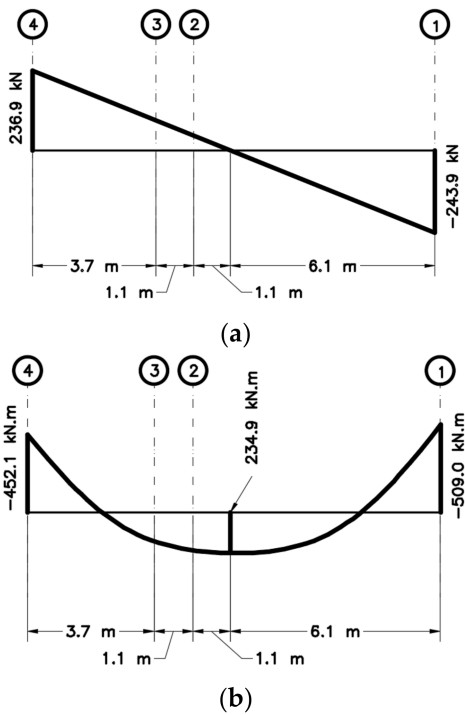

**(a)**

**(b)**

**Figure 12.** Efforts in beam V3 to the initial time instant $t = 0$: (**a**) shear force (kN); (**b**) bending moment (kN m).

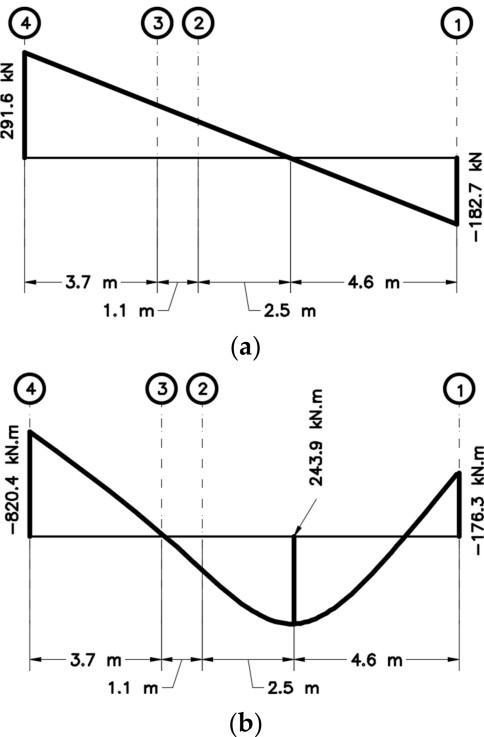

**(a)**

**(b)**

**Figure 13.** Efforts in beam V3 to the infinite time, after iterative process: (**a**) shear force (kN); (**b**) bending moment (kN m).

The changes in the bending moment and in the shear force at one end of the beam V3 can be explained by the increase in stiffness of the S7 footing in relation to the S1 footing, based on the second and fifth iterations of Table 2.

## 5. Conclusions

A numeric procedure for predicting consolidation settlements by the Kelvin–Voigt model was presented, applying the compatibility of the displacements to the structure of the building, to the structure of the footings, and to the mass of soil through the Finite Element Method coupled with the Boundary Element Method and Terzaghi's Theory of Consolidation. The presentation is shown with an explicit application on a study case. The method can be used for any other structural system under footing foundations sensitive to consolidation settlements.

The BEM has been showing an interesting solution to model elastic and viscoelastic media [44]. It is also possible to state that here the computational cost was shown to be considerably lower, comparing the considered methodology with the conventional soil modeled by finite elements [53].

The building analyzed here is a case of consolidation settlement. The required forces obtained for some footings, beams, and columns, had relevant increases, after a redistribution of efforts. The relative rotations observed in the building under study resulted in levels higher than the recommended value for preventing some service limit state. The S1 = S6 and S7 = S12 footings showed a load value that exceeds 35% of the initial value at infinite time and their allowed stresses is violated over time. The methodology also shows it is efficient to plan the deadline for structural interventions in foundations. The beams V2 and V3 of the first floor presented an increase in shear force and in the bending moment. Column P7 presented a compression increase in the order of 39%, compared to the initial value, and the flexural efforts increase significantly in the XZ and YZ planes. Therefore, it is concluded that for the structure to be able to accommodate at predicted displacement levels, a superstructure would need to be previously dimensioned to tolerate additions of efforts resulting from slow accommodation movements or to have its structural elements reinforced, where it is necessary.

**Author Contributions:** Conceptualization, R.M.L., M.G. and V.d.S.A.; methodology, R.M.L., M.G. and V.d.S.A.; software, R.M.L. and V.d.S.A.; validation, R.M.L. and V.d.S.A.; investigation, R.M.L.; resources, R.M.L. and V.d.S.A.; data curation, R.M.L. and M.G.; writing—original draft preparation, R.M.L. and M.G.; writing—review and editing, R.M.L., M.G. and V.d.S.A.; supervision, M.G.; project administration, M.G.; funding acquisition, M.G. All authors have read and agreed to the published version of the manuscript.

**Funding:** This research was funded by Fundação de Amparo à Pesquisa do Estado de Minas Gerais (FAPEMIG), grant number PPM-00444-18, and Conselho Nacional de Desenvolvimento Científico e Tecnológico (CNPq), grant numbers 405183/2018-6, 302597/2019-0, 302119/2022-1.

**Data Availability Statement:** All data used and materials developed in this research are available by request.

**Conflicts of Interest:** The authors declare no conflict of interest.

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
