# Peer review of "Viscoelastic Soil–Structure Interaction Procedure for Building on Footing Foundations Considering Consolidation Settlements"

_buildings, doi:10.3390/buildings13030813_

Round 1

Reviewer 1 Report (Previous Reviewer 2)

The authors have submitted the revised manuscript. Although I am mostly satisfied with their revisions.

Author Response

Point 1: The authors have submitted the revised manuscript. Although I am mostly satisfied with their revisions.

Response 1:

The authors are grateful for contributions.

Reviewer 2 Report (Previous Reviewer 3)

The authors still fail to support why boundary elements are required to address the problem at hand, namely the consolidation settlement of foundations. The problem, at first sight, involves a limited domain as long the drainage boundaries are clearly stated. Boundary elements are most useful when dealing with large or even infinite domains, which is not the case, at first sight, for this problem. 

Moreover, the authors try to develop a botched assessment of the performance of their approach by referring to a publication that addresses a completely different problem (earthquake response of a building using a non-linear model), which is not proper. 

Author Response

Point 1: The authors still fail to support why boundary elements are required to address the problem at hand, namely the consolidation settlement of foundations. The problem, at first sight, involves a limited domain as long the drainage boundaries are clearly stated. Boundary elements are most useful when dealing with large or even infinite domains, which is not the case, at first sight, for this problem. Moreover, the authors try to develop a botched assessment of the performance of their approach by referring to a publication that addresses a completely different problem (earthquake response of a building using a non-linear model), which is not proper.

Response 1:

The problem effectively really does not involve a limited domain. The superstructure is supported by multiple footings (both limited domains) that are directly connected by the soil that is an intrinsic semi infinite problem. Any correlated phenomenon elastically coupled in this system, even a local or with boundaries that are clearly stated, generates elastic influence on the whole mass, that is governed, as known, by the fourth order partial differential equations of three Galerkin functions that must be attended by infinite boundaries. Most of the problem in geomechanics has an extensive list of references by applying Boundary Element Method (BEM), as it seen in Witt [2], Davies and Banerjee [3], and Ai et al. [13-14, 20].

In fact, the drainage boundaries are clearly stated but its phenomenon involves the changing of the stress and pore pressure responses at long of the consolidation time. This is expressed by the equations (1) or (2) that has a strong influence in the equation (10), where the elastic and viscous stresses are altered at each time step, and finally in the BEM solution, obtained by the discretization of the equation (24), with the vertical displacements are obtained iteratively (at each time). All this analysis has naturally been contemplated by the group effects among all the footings. As the settlements have changed in a certain time of the gradual reduction in volume of the mass, obtained by semi-infinite elastic analysis, the efforts in the building and reactions in the multiple footings have been changed naturally.

Simple methods could be applied in the aforementioned analysis, like the Winkler model or the two-parameter models (Pasternak or  Filonenko–Borodich model), but for general engineering applications, like presented in the paper, these simple models are limited. For instance, they usually do not take into account adequately the group effect of the multiple footings and the difficulty of choosing the mechanical parameters for the stratum. 
In this view, the numerical procedures like the use of Finite Element (FEM), Finite Difference (FDM) or Boundary Element (BEM) methods have been applied extensively for more than 40 years for solving three-dimensional continuity medium. As written by Schanz [44] about the BEM: “The main advantages of the method are reduction of the problem dimension by one and the implicit fulfillment of the radiation condition for unbounded domains. Due to the last advantage the BEM seems to be preferable for the calculation of infinite or semi-infinite domains, e.g. soil”. 

Finally, the consideration of the physical phenomenon of the gradual reduction in the volume of the fully saturated soil, consolidation theory, does not disqualify the application of the infinite domain in the present analysis, on the contrary, as pointed out in several important works using BEM [13-14, 20, 44]. The application of the BEM for solving the semi-infinite medium considering the consolidation theory coupled with the FEM for the analysis of the limited domain: multiple footings and the building with non linearity; is much more necessary due to the low storage and computational cost for solving the whole complex problem effectively and with more accuracy.

References

[2] Witt M. Solutions of plates on a heterogeneous elastic foundation. Computers & Structures, 1984, 18, 41-45. https://doi.org/10.1016/0045-7949(84)90080-4

[3] Davies, T.G; Banerjee, P.K. The displacement field due to a point load at the interface of a two layer elastic half-space. Géotechnique, 1978, 28, 43-56. https://doi.org/10.1680/geot.1978.28.1.43

[13] Ai, Z. Y.; Chu, Z. H.; Cheng, Y. C. Time-dependent interaction between superstructure, raft and layered cross-anisotropic viscoelastic saturated soils. Applied Mathematical Modelling, v. 89, p. 333-347, 2021. https://doi.org/10.1016/j.apm.2020.07.018

[14] Ai, Z.Y.; Cheng, Y.C.; Cao, G.J. A quasistatic analysis of a plate on consolidating layered soils by analytical layer‐element/finite element method coupling. International Journal for Numerical and Analytical Methods in Geomechanics, 2014, 38. 1362-1380. https://doi.org/10.1002/nag.2261

[20] Ai, Z. Y.; Jiang, Y. H.; Zhao, Y. Z.; Mu, J. J. Time-dependent performance of ribbed plates on multi-layered fractional viscoelastic cross-anisotropic saturated soils. Engineering Analysis with Boundary Elements, v. 137, p. 1-15, 2022. https://doi.org/10.1016/j.enganabound.2022.01.006.

[44] Schanz, M. A boundary element formulation in time domain for viscoelastic solids. Communications in Numerical Methods in Engineering,15, n. 11, p. 799-809, 1999. https://doi.org/10.1002/(SICI)1099-0887(199911)15:11<799::AID-CNM294>3.0.CO;2-F

Reviewer 3 Report (New Reviewer)

The manuscript presents an interesting case study about viscoelastic soil-structure interaction for building on footing foundations considering consolidation settlements. Nevertheless, the novelty of the study is not clear, and conclusions seem to be restricted to “an extreme case of severe consolidation settlement” as it is stated in the conclusion section. Therefore, it is recommended to improve the paper by discussing the state of the art and what is the novelty of this research.

Other comments:

The introduction section should be considerably improved. The state of the art is not presented in the introduction section. Moreover, it should be clarified what is the novelty of this study.

Page 3, line 114. It is not clear what you mean by saying that soil is deformable and interstitial fluid incomprehensible. By the soil, do you mean the soil grains or the whole soil? In this case, the compression of the soil layer is due to the change in volume only, with is due to the squeezing out of the water from the void spaces (compressibility of either water or soil grains is negligible).

Page 3, line 121. It is stated for unsaturated soils that immediate settlements are deformations at constant volume (no change in void ratio, only change in shape), being dominant in non-cohesive soils. The immediate settlement without change in void ratio is usually attributed to saturated cohesive soils. For non-cohesive soils, a reduction in the void ratio is expected. Please, rewrite the sentence or use references to support it.

Subsection 3.2 starts with “In that paper,…”. It is recommended to cite again the paper as a new subsection begins.

Author Response

Point 1: The manuscript presents an interesting case study about viscoelastic soil-structure interaction for building on footing foundations considering consolidation settlements. Nevertheless, the novelty of the study is not clear, and conclusions seem to be restricted to “an extreme case of severe consolidation settlement” as it is stated in the conclusion section. Therefore, it is recommended to improve the paper by discussing the state of the art and what is the novelty of this research.

Response 1:

In Chapter 1 (introduction), a general review of the text was carried out. The state of the art was discussed, with the inclusion of some current references, and an explicitly description of research innovations was shown.

In Chapter 5 (conclusions), the text was improved and the term "extreme" was removed.

The authors are grateful for contributions.

Point 2: Other comments: The introduction section should be considerably improved. The state of the art is not presented in the introduction section. Moreover, it should be clarified what is the novelty of this study.

Response 2:

In Chapter 1 (introduction), a general review of the text was carried out. The state of the art was discussed, with the inclusion of some current references, and an explicitly description of research innovations was shown.

The authors are grateful for contributions.

Point 3:  Page 3, line 114. It is not clear what you mean by saying that soil is deformable and interstitial fluid incomprehensible. By the soil, do you mean the soil grains or the whole soil? In this case, the compression of the soil layer is due to the change in volume only, with is due to the squeezing out of the water from the void spaces (compressibility of either water or soil grains is negligible).

Response 3:

The sentence has been improved in the text.

The authors are grateful for contributions.

Point 4:  Page 3, line 121. It is stated for unsaturated soils that immediate settlements are deformations at constant volume (no change in void ratio, only change in shape), being dominant in non-cohesive soils. The immediate settlement without change in void ratio is usually attributed to saturated cohesive soils. For non-cohesive soils, a reduction in the void ratio is expected. Please, rewrite the sentence or use references to support it.

Response 4:

The sentence has been removed from the text.

The authors are grateful for contributions.

Point 5: Subsection 3.2 starts with “In that paper,…”. It is recommended to cite again the paper as a new subsection begins.

Response 5:

The citation was carried out using “Here” to refer to own paper.

The authors are grateful for contributions.

This manuscript is a resubmission of an earlier submission. The following is a list of the peer review reports and author responses from that submission.

Round 1

Reviewer 1 Report

This paper investigates the frame structures supported on footing foundations subjected to slow strains caused by consolidation settlements. The topic related to the SSI and the aim of the paper is within the journal. I believe that the contributions of this research work are clearly stated. Notwithstanding, a few aspects of the present submission need further clarifications and integrations through a minor revision, as specified below:

(1) The adopted software should be explained. 

(2) The font style and figure presentation of Figs. 8-12 should be unified and improved. 

(3) The new finding compared to the immediate displacement analysis should be further highlighted in the Conclusion.

Reviewer 2 Report

MAJOR CONCERNS

1. The  main theme of the manuscript is to demonstrate that Kelvin-Voigt viscoelastic model can be successfully used to predict total settlements and the rate of settlements for shallow foundations. The authors have used only one settlement data point (at 10 years) of a building (fictitious building as they describe it, Lines 207 of the manuscript) on shallow foundations to estimate the viscoelastic parameters of a 10 meter compressible soil. I am not sure what is meant by fictitious. Does the building exist? Is the data fictitious? The estimated viscoelastic parameters based on only one point are not reliable and cannot be used for extrapolation. 

2. Section 2 of the manuscript includes a presentation of commonly known information available in textbooks. The information in this section can be summarized in a few sentences or short paragraphs just to state the methodology that the authors followed.

3. Section 3.1, Problem Description, First three paragraphs. The authors mention two buildings: one is 26 stories existing building (10 mm/year settlement rate), and the other 19 stories "fictitious" hotel building (15 mm/year settlement rate). What is the connection between these two? Are these settlement rates constant over the 10 years? Were the design, loads, and the settlements of the "fictitious" hotel made up or assumed? 

4. Lines 230 to 233 - No laboratory data nor any reference is provided. What is the basis of the estimate of the coefficient of consolidation value ? If the settlement rate is constant at 15 mm/year as the authors suggest, then the estimated value of the coefficient of consolidation is not reliable. Consequently, the settlements in figure 4 cannot be verified. The only corrected data point in Figure 4 is the settlement at 10 years.

5. Table 1 gives the AVERAGE stresses that are used in the calculations. This means that 50% of the time the foundations experience stresses larger than the AVERAGE stresses. What is the justification for using the AVERAGES?

6. The authors need to include in the literature review and the references relevant citations to the research topic. Very few references mention the word VISCOELASTIC.  

MINOR CONCERNS

The following is a sample of the numerous concerns in the manuscript:

1. Line 37 - What is "dense study"?

2. Lines 40 to 42 - This sentence does not make sense.

3. The authors use "velocity" (Lines 27, 37 and 122 , a vector), and "speed" (Lines 203 and 215, a scaler), interchangeably. I believe the term "speed" is more appropriate to use in the context of the manuscript.

Reviewer 3 Report

You address a topic that is relevant to both industry and research. However, the reasons why you need to use boundary elements are not clearly presented. You are solving a consolidation analysis for a large building, which can be assessed in a straightforward manner in a limited domain. This can be done quite efficiently using finite element models, even allowing for inelastic behaviour and considering the flow of water. You can do this with Opensees. 

The great advantage of Boundary elements is the fact that assessments can be done on the boundary of the problem, so large domains can be studied efficiently, but that is not the type of problem you are addressing. Boundary elements are better suited for vibration problems, where you have large domains and you have strains and displacements away from the region of interest. Moreover, your increased complexity is self-defeating as you can't capture efficiently changes in strain-stress behaviour within the domain. This is a significant shortcoming of your assessment, as you seem to enforce elastic behaviour, but using a more complex tool, than regular FEM analyses.